# Natural History of Aerosol Induced Lassa Fever in Non-Human Primates

**DOI:** 10.3390/v12060593

**Published:** 2020-05-29

**Authors:** Isaac L. Downs, Carl I. Shaia, Xiankun Zeng, Joshua C. Johnson, Lisa Hensley, David L. Saunders, Franco Rossi, Kathleen A. Cashman, Heather L. Esham, Melissa K. Gregory, William D. Pratt, John C. Trefry, Kyle A. Everson, Charles B. Larcom, Arthur C. Okwesili, Anthony P. Cardile, Anna Honko

**Affiliations:** 1US Army Medical Research Institute of Infectious Diseases, Fort Detrick, Frederick, MD 21702, USA; isaac.l.downs.mil@mail.mil (I.L.D.); carlshaia@gmail.com (C.I.S.); xiankun.zeng.ctr@mail.mil (X.Z.); joshua.johnson@nih.gov (J.C.J.); lisa.hensley@nih.gov (L.H.); david.l.saunders.mil@mail.mil (D.L.S.); franco.d.rossi.ctr@mail.mil (F.R.); kathleen.a.cashman.ctr@mail.mil (K.A.C.); heather.l.esham.civ@mail.mil (H.L.E.); melissa.k.gregory3.ctr@mail.mil (M.K.G.); hoomnong@verizon.net (W.D.P.); john.c.trefry.civ@mail.mil (J.C.T.); kyle.a.everson.mil@mail.mil (K.A.E.); arthur.c.okwesili.mil@mail.mil (A.C.O.); honko@bu.edu (A.H.); 2Integrated Research Facility, National Institute of Allergy and Infectious Diseases, National Institutes of Health, Frederick, MD 21702, USA; 3Defense Threat Reduction Agency, Fort Belvoir, VA 22060, USA; 4Madigan Army Medical Center, Joint Base Lewis-McChord, WA 98431, USA; charles.b.larcom2.mil@mail.mil; 5Investigator at National Emerging Infectious Diseases Laboratories, Boston University School of Medicine, Boston, MA 02118, USA

**Keywords:** LASV, Lassa, Josiah, VHF, telemetry, aerosol, anion gap, alkalosis, macaque, virus

## Abstract

Lassa virus (LASV), an arenavirus causing Lassa fever, is endemic to West Africa with up to 300,000 cases and between 5000 and 10,000 deaths per year. Rarely seen in the United States, Lassa virus is a CDC category A biological agent inasmuch deliberate aerosol exposure can have high mortality rates compared to naturally acquired infection. With the need for an animal model, specific countermeasures remain elusive as there is no FDA-approved vaccine. This natural history of aerosolized Lassa virus exposure in *Macaca fascicularis* was studied under continuous telemetric surveillance. The macaque response to challenge was largely analogous to severe human disease with fever, tachycardia, hypotension, and tachypnea. During initial observations, an increase trend of activated monocytes positive for viral glycoprotein was accompanied by lymphocytopenia. Disease uniformly progressed to high viremia followed by low anion gap, alkalosis, anemia, and thrombocytopenia. Hypoproteinemia occurred late in infection followed by increased levels of white blood cells, cytokines, chemokines, and biochemical markers of liver injury. Viral nucleic acids were detected in tissues of three non-survivors at endpoint, but not in the lone survivor. This study provides useful details to benchmark a pivotal model of Lassa fever in support of medical countermeasure development for both endemic disease and traditional biodefense purposes.

## 1. Introduction

Lassa virus (LASV) is the causative agent for Lassa fever, which is a viral hemorrhagic disease endemic to Western Africa. An average of 100,000–300,000 cases of the naturally occurring disease exist annually with approximately 5000–10,000 mortalities [1,2,3,4]. Mortality rates for those hospitalized for Lassa fever are 15%–50% [5,6]. LASV is on the United States Federal Select Agents and Toxins List posing a severe threat to human and animal health [7]. It is also classified as a CDC Category A biological agent and Risk Group 4 capable of causing serious disease, transmissible from person-to-person, and does not have definitive prevention or treatment [6,8,9,10].

*Mastomys natalensis*, a multimammate rodent is thought to be the primary natural reservoir of LASV [11,12]. The classical mechanism of infection is through direct contact with the animal itself, contact/inhalation of excreta, or ingestion of contaminated food [11]. Multiple human-to-human transmissions have been reported for imported LASV in non-endemic areas [13,14,15,16]. Major concerns for imported Lassa fever cases are severely infected commercial air travelers and the international transport of infected *Mastomys natalensis*. Given the availability of the natural reservoir host in the endemic area, LASV is a biological threat due to its potentially high mortality rate and ease of transmission.

The stability of aerosolized Lassa virus is influenced by environmental factors such as relative humidity and temperature. The maximum half-life of LASV is 54.6 min at 24 °C with relative humidity of 30%, similar to dry season conditions in West Africa [17]. As described in Stephenson et al. in 1984, all nine *Macaca fascicularis* (cynomolgus macaques) succumbed to disease after inhaling doses of Lassa virus Josiah strain (NCBI:txid11622) [17]. Though fever was detected on day 4–6 post exposure, the effects of environmental factors on virus stability and aerosol infectivity were investigated contrary to signs of disease. Thereby, a major gap exists in the clinical knowledge of aerosol-induced Lassa fever.

Previous studies of cynomolgus macaques have shown a high degree of similarity with the human disease when intramuscularly or subcutaneously challenged with LASV [18,19,20,21,22,23]. Several prior studies intermittently used anesthetics to humanely collect blood samples from non-human primates (NHPs); a process currently recognized to potentially interfere with immune responses [24,25]. Accordingly, the FDA states that “natural history studies are studies in which animals are exposed to a challenge agent and monitored to gain an understanding of the development and progression of the resulting disease or condition,… the time from exposure to manifestation onset, time course and order of manifestation progression, and severity” [26]. This NHP study establishes the natural history of Lassa fever during aerosol exposure using venous catheterization and surgically implanted radio telemetry devices to monitor and collect physiological parameters. The study was performed during the adoption of state-of-the-art technology in order to safely investigate and characterize the evolution of the disease in real time.

## 2. Materials and Methods

### 2.1. Agent Preparation and Challenge Conditions

Research was conducted in compliance with the Animal Welfare Act and other federal statutes and regulations relating to animals and experiments involving animals. Protocols approved by Institutional Animal Care and Use Committee adhered to principles stated in The Guide for the Care and Use of Laboratory Animals, National Research Council, 1996. The research facility where experiments where performed is fully accredited by the Association for Assessment and Accreditation of Laboratory Animal Care, International.

LASV Josiah strain was selected as the challenge Lassa virus due to its well-documented history in the cynomolgus model of Lassa fever [18,19,20,21,22,23,27]. As previously performed, target dose of 1000 plague forming units (PFU) of aerosol LASV were performed in a class III biosafety cabinet inside a BSL-4 laboratory using a 3-jet collision nebulizer (BGI Inc., Waltham, MA, USA) and with the head only automated bioaerosol exposure II (ABES-II) system developed at USAMRIID [28,29]. Environmental conditions of head compartment were generally 26.2 °C with 52% relative humidity. NHPs were anesthetized with Telazol^®^ at 3 mg/kg (Fort Dodge Animal Health, New York, NY, USA) via central venous catheter (CVC) and immediately after the aerosol exposure, the NHPs were evaluated in a whole body plethysmography box (Buxco Research Systems, Wilmington, NC, USA) in order to determine each NHP’s minute volume, from which individual challenge doses were calculated.

### 2.2. Animals and Telemetry

Four male cynomolgus macaques (*Macaca fascicularis*) between 8.2 and 8.9 kg from Worldwide Primates, Inc. (Miami, FL, USA) were chosen for the study. Each NHP was implanted with a T27 Integrated Telemetry Systems (ITS) radio telemetry device (Konigsberg Instruments, Inc., Pasadena, CA, USA) in order to accurately gather physiological data in real time without potentially compromising the course of disease with anesthetics and stressful animal manipulations. Following the ITS implantation surgery, NHPs were allowed to recover for 6 months prior to Groshong 7F CVC (Bard Access Systems, Salt Lake City, UT, USA) implantation for the purpose of repeated blood sample collection. NHPs were fitted with Lomir jackets (Malone, NY, USA) in order to protect the catheters and allow the NHPs full range of movement within their enclosures. NHPs were left to recover for at least one week after CVC surgery prior to their transfer into the BSL-4 and additional one-week acclimation period before challenge. Body temperature, arterial blood pressure, left ventricular pressure, intrathoracic pressure to derive respiratory rate, and electrocardiogram were continuously recorded, and data averaged over thirty minute intervals for each NHP throughout the duration of the study.

### 2.3. Daily Observations

All animals were observed at least twice daily from three days prior to challenge until the endpoint of the study. These observations were performed at least four hours apart, and scores are derived from detailed descriptions in the clinical assessment in accordance to institutional guidelines. Score codes were generally normal (0), very mild (1), mild (2), moderate (3), and severe (4) for responsiveness/behavior, rash, bleeding, gastrointestinal (GI) symptoms, posture/appearance, and respiration. Score codes were also normal (0), mild (1), moderate (2), and severe (3) for food consumption, urine output, and edema. Vocalization assessment include normal (0) and in distress (1). Exudate assessment includes normal (0) and present (1). Animals were euthanized under deep anesthesia when evaluation criteria reached a number in correlation to historical control data where 100% animals were humanely euthanized.

### 2.4. Blood Sampling

Prior to drawing blood samples, the CVC was flushed with saline from a 10 mL PosiFlush saline syringe (BD Biosciences, San Jose, CA, USA). Blood samples were collected daily from each NHP’s CVC between days 3 through 16 post-exposure with a 5 mL syringe and aliquoted into 0.5 mL K2EDTA MAP tubes (BD Biosciences), 1 mL serum with clot activator tubes, and 1 mL sodium citrate MiniCollect tubes (Greiner Bio-One, Monroe, NC, USA). After day 16, blood was collected from the survivor on day 30 and 41. The CVC line was again flushed with saline and maintained by flushing with PosiFlush pre-filled heparin lock syringes (BD Biosciences), which were left attached to the CVC until the next sample collection. On days where sampling via CVC was complicated due to kinking of the line, NHPs were anesthetized with ketamine HCl at 10 mg/kg (VedCo, Saint Joseph, MO, USA) or a combination of Tiletamine HCl and Zolazepam HCl at 3 mg/kg (Fort Dodge Animal Health, New York, NY, USA). Survivor and D10:Non-survivor had catheter kinking three days prior to challenge; no other kinks are noted. Once anesthetized, blood was drawn via saphenous venipuncture.

### 2.5. Blood Gases

After blood samples were aliquoted from the syringe into the various MiniCollect tubes, approximately 150 µL of untreated, uncoagulated whole blood were used to fill the sample ports of a CG4+ blood gas/lactate/pH analysis cartridge, CHEM8+ cartridge, and a PT/INR prothrombin time test cartridge for use in an i-STAT handheld analyzer (Abbott Laboratories, Abbott Park, IL, USA).

### 2.6. Blood Chemistry

At each time point, serum supernatant was collected and applied to a Piccolo General Chemistry 13 reagent disc for subsequent analysis in a Piccolo point-of-care blood chemistry analyzer (Abaxis, Union City, CA, USA) as previously performed [20,28]. Serum was evaluated for alkaline phosphatase (ALP), alanine aminotransferase [13], aspartate aminotransferase (AST), blood urea nitrogen (BUN), creatinine (CRE), and gamma glutamyltransferase (GGT) levels. Plasma was applied to general chemistry reagent disks to evaluate albumin, hemolysis, total bilirubin, total protein, and uric acid levels. As described above in blood gases, CHEM8+ cartridges were used to evaluate anion gap, chloride, glucose, ionized calcium, potassium, and sodium.

### 2.7. Cytokine and Chemokine Analysis

Circulating cytokines and chemokines were analyzed using a Milli-Plex MAP NHP pre-mixed 23-plex assay (Millipore) as previously performed [28].

### 2.8. Flow Cytometry

Fluorochrome-conjugated monoclonal antibodies cocktails (2× concentration) were prepared using anti CD45 PerCP, CD3 V450, CD4 QD655, CD8 QD-705, CD20 PE/AlexaFluor 700, CD14 FITC, CD16 APC-H7, αLASV-GP QD605, or IgG1 QD605 [30]. Then, 50 µL of the prepared antibody cocktail was added to 50 µL whole blood within a deep well plate (Nunc). Plates were incubated in the dark at room temperature for 15 min. Each well with whole blood/antibody cocktail solution received 850 µL of FACS lysing solution (BD Biosciences), then plates were incubated for an additional 15 min. After the incubation period, plates were centrifuged at 450× *g* for 5 min to pellet the cells and then inverted to remove the liquid. Cells were resuspended in 850 µL BSA FACS Staining buffer (BD Biosciences) and centrifuged at 450× *g* for 5 min to pellet. Plates were again inverted to remove the supernatant, then cells were resuspended in 300 µL BSA FASC Staining buffer and transferred to 5 mL FACS tubes (BD Biosciences) for flow cytometry analysis. Appropriate staining controls were prepared according to manufacturer’s instructions (BD Biosciences). Compensations were set according to manufacturer’s instructions for each of the stains used in the assay. Samples were then run on a LSR II Fortessa flow cytometry instrument and data was acquired. In accordance with immunophenotypic characteristics from Gorczyca et al. [31], gating analyses were performed with IgG1 or αLASV-GP for granulocytes (CD45+ SSC-A^hi^ FSC-A^hi^), classical monocytes (CD45+ SSC-A^lo-mid^ FSC-A^mid^ CD3- CD20- CD14+ CD16-), activated monocytes (CD45+ SSC-A^lo-mid^ FSC-A^mid^ CD3- CD20- (CD14+ CD16+ and CD14-CD16+)), B cells (CD45+ SSC-A^lo-mid^ FSC-A^mid^ CD20+ AF700 D0-D4 and PE D7), T cells (CD45+ SSC-A^lo-mid^ FSC-A^mid^ CD3+), CD4+ T cells (CD45+ SSC-A^lo-mid^ FSC-A^mid^ CD3+ CD4+), and CD8+ T cells (CD45+ SSC-A^lo-mid^ FSC-A^mid^ CD3+ CD8+).

### 2.9. Hematology

Blood samples in 0.5 mL K2EDTA MAP tubes were inverted 8–10 times. Directly after inversion, the tube was fitted to the probe of a Hemavet 950FS hematology analyzer (Drew Scientific Inc., Dallas, TX, USA) to generate a complete blood count (CBC) profile for each NHP at specified time points. Prior to analysis, proper Hemavet operation was verified by CBC control from the manufacturer (Drew Scientific Inc.). When parameters were within the manufacturer’s acceptability, CBCs were performed on each of the blood samples. If the control values were not in the acceptable range, the machine was recalibrated based on the values obtained from the CBC controls.

### 2.10. Immunohistochemistry

Immunohistochemistry (IHC) was performed using Dako Envision system (Dako Agilent Pathology Solutions) as previously reported [19]. Briefly, after deparaffinization, rehydration, and methanol/hydrogen peroxide blocking, slides were stained using a mouse anti-LASV GP2 (USAMRIID, 52-2074-7A) at a dilution of 1:8000, followed by an HRP-conjugated, secondary anti-mouse antibody. All slides were exposed to brown chromogenic substrate DAB, counterstained with hematoxylin, dehydrated, and cover slipped.

### 2.11. In situ Hybridization

*In situ* hybridization was performed to detect LASV genomic RNA in histological tissues using the RNAscope 2.5 HD RED kit (Advanced Cell Diagnostics) according to the manufacturer’s instructions as previously described [19].

### 2.12. Pathology

Complete necropsies were performed at the time each NHP was euthanized or succumbed to Lassa infection. Samples of all major organs and tissues were collected, then fixed in 10% formalin for histology and immunohistochemistry (IHC) as described previously [19].

### 2.13. Plaque Assays

Virus titration was performed via plaque assay on Vero E6 cells (ATCC, Manassas, VA, USA) grown in 10% fetal bovine serum (FBS) supplemented α-MEM (Gibco). Cells were plated in sterile 6-well plates and utilized once they reached >90% confluence; at this time, the medium was removed from each well and replaced with 200 µL of the appropriate material or dilution to be titered. Viral inoculums were incubated on the monolayer for 1 h at 37 °C and 5% CO2 in a humidified incubator. After 1 h the monolayers were overlaid with the micro-crystalline methyl cellulose semi-solid Avicel-591 (FMC Biopolymer, Philadelphia, PA, USA) for a 1.25% final concentration in growth medium and incubated for 8–9 days at 37 °C and 5% CO2 in a humidified incubator. After the 8–9-day incubation, a solution of 0.4% crystal violet in neutral buffered formalin was added to each well and incubated overnight. Wells were washed with water, plaques counted, and titers calculated by multiplying the averaged plaque counts by the dilution factor for each condition. Titers are expressed in terms of plaque forming units (PFU) per milliliter.

### 2.14. qRT-PCR

For RNA extraction, plasma was thawed and centrifuged at 6800× *g* for 3–5 min to clear cryoprecipitates, after which 200 µL of supernatant citrated plasma was added to 600 µL of Tri Reagent LS (Sigma, St. Louis, MO, USA). Following decontamination and release from the biocontainment suite, samples were stored at −80 ± 10 °C until processed. Nucleic acids were separated from protein via 2 mL phase lock gel tubes (5Prime, Gaithersburg, MD, USA) and RNA was isolated using the Qiagen RNeasy Mini kit (Qiagen, Germantown, MD, USA). Aliquots were frozen at −80 ± 10 °C until analysis. One-step quantitative real-time qRT-PCR reactions were performed on a LightCycler 480 (Roche, Indianapolis, IN, USA) in 20 μL volumes made of 5 μL of purified RNA and 15 µL of mastermix containing the RNA SuperScript II One-Step RT-PCR System (Life Technologies).

PCR amplification of the LASV glycoprotein gene from the S segment was performed as previously cited [28]. Absolute quantification of viral gene expression was based on the creation of a viral standard curve through 10-fold serial dilution of the stock virus in the challenge matrix and subsequent extraction of RNA. The LightCycler 480 software (version 1.50) was then used to relate viral genomes to plaque forming units through the standard curve values compared to standard titers of the stock virus from the plaque assay.

## 3. Results

### 3.1. Clinical Observations

The natural history study of aerosolized high-dose Lassa virus was evaluated by exposing four healthy cynomolgus macaques to a target dose of 1000 PFU of LASV-Josiah strain by the aerosol exposure route. No signs of illness were observed in the subclinical stage of the disease from day-post exposure (DPE) 0–3. Between DPE 4–7, all animals exhibited clinical score of disease or detectable fever and viremia (Figure 1, Appendix A). All animals had changes in posture and appearance with impaired neurological function occurring within DPE 8–23 (Appendix A) and transient increase in heart rate and temperature on DPE 25–29 relative to duration of survival. This timeframe is noted as the decompensated stage of the disease.

In accordance to protocol, two macaques were euthanized when displaying severely subdued behavior and shaking on DPE 10. An animal euthanized on DPE 13, known as D13:non-survivor, experienced epistaxis on DPE 12 as well as facial edema and petechiae on DPE 13. The challenge dose was fatal for three of four exposed macaques with body weight loss observed in D10:non-survivor (1.9 kg), D13:non-survivor (3.9 kg), and D10:non-survivor.2 (1.3 kg). All non-survivors had dry, white fecal matter on DPE 5. The survivor uniquely had epistaxis on DPE 6 as well as watery diarrhea on DPE 8 that later subsided on DPE 23. The highest mean clinical score of the survivor occurred on DPE 13 (Appendix A). The survivor’s average assessment score of posture and appearance as well as neurological function score was mild (2) compared to moderate endpoint scores for D10:non-survivor (2.5), D13:non-survivor (2.5), and D10:non-survivor.2 (3.5). No signs of illness were identified in post-DPE 30 assessments of the single survivor to which body weight loss (2.5 kg) was determined when euthanized on DPE 41 (Appendix A).

### 3.2. Telemetry

All NHPs were implanted with radio devices to monitor physiologic data in real time to prevent intermittent use of anesthetics and their stressful manipulations. Three days prior to viral challenge, baseline readings of animals indicate diurnal variations of temperature, heart rate, electrocardiogram detection of QRS complex duration, respiratory rate, and mean arterial pressure (Appendix A). All infected animals had a fever, greater than 1.5 °C above baseline for longer than 2 h, by DPE 4 (Figure 2a). A constant fever in NHPs was evident by DPE 8. The survivor had the highest mean daily temperature and sustained the longest fever of 4 days contrary to variable durations observed with non-survivors. Within DPE 9–13, these non-survivors experienced hypothermia based on temperatures less than 2 °C from baseline for longer than 2 h (Appendix A). The temperature readings from the survivor (Appendix A) return to baseline levels on DPE 16, and then mildly increase on DPE 25–41 (Appendix A).

Aerosol LASV-exposed NHPs displayed tachycardia (Figure 2b), tachypnea (Figure 2c), and hypotension (Figure 2d) within DPE 8–14 [32] (pp. 49). Changes in heart rate (Appendix A) and QRS complex duration (Appendix A) are linked to increased physical activity with no indication of abnormal ventricular depolarization. This decline in QRS complex duration did not occur in all animals, such as survivor on DPE 4–14. There were no clinically significant arrhythmias noted during the course of disease preceding euthanasia (Appendix A). The survivor had premature ventricular complexes (PVCs) at baseline (Appendix A) and the D13:non-survivor displayed PVCs late in the course of disease (Appendix A). Interestingly, 3 out of 4 NHPs had subtle PR depressions, and 2 of 4 NHPs had decreased QRS complex voltage during the course of disease (Appendix A).

Tachypnea was observed in survivor within DPE 4–14 (Appendix A) as well as non-survivors from DPE 8 until euthanasia (Appendix A) [32] (pp. 49). Hypotension was observed in all animals between DPE 8–20 (Figure 2d, Appendix A) based on mean arterial blood pressure values significantly less than 3.0 standard deviations (LS) [32] (pp. 49). These physiologic parameters of infected NHPs can be divided into three stages: (1) subclinical stage prior to fever (DPE 0–3), (2) the clinical stage with fever and tachycardia (DPE 4–7), and (3) decompensated stage with marked hypothermia, tachycardia, tachypnea, and hypotension (DPE 8–14).

### 3.3. Viremia

Viremia titers of plasma were trended using standard plaque assays as described in materials and methods. Plasma viremia was first observed with survivor macaque (● Survivor) on DPE 3 followed by all non-survivors on DPE 4 that later succumb to the disease (Figure 3). Peak viremia levels (1.81–8.00 × 10^6^ PFU/mL) occurred in three non-survivors on DPE 10–13 at termination. The survivor had peak viremia levels (8.91 × 10^5^ PFU/mL) on DPE 13; lower level viremia on DPE 30 (8.27 × 10^1^ PFU/mL) followed no detectable viremia on day 41.

### 3.4. Blood Gases and Clinical Chemistry

After aerosol exposure to Lassa virus, variations in venous gases were observed during the clinical and decompensated stages of the disease (Figure 4a–e, Appendix A). Venous partial pressure of carbon dioxide decreased 17.9% (pCO_2_: Figure 4a) and total carbon dioxide decreased 9.9% (tCO_2_: Figure 4b) in non-survivors on DPE 5, which return near baseline levels on DPE 10 [33]. However, the survivor did not experience decreased venous pCO_2_ and tCO_2_ levels like non-survivors. High venous pH levels, greater than 7.5, were observed in all animals on DPE 8 (Figure 4c) [33]. After DPE 8, venous pH levels return near baseline and became acidic on the day of euthanasia. Bicarbonate levels decreased 9.7% in non-survivors on DPE 5 (Figure 4d). The low venous pCO_2_ and high pH levels indicate respiratory alkalosis occurred in non-survivors followed by expected compensation via the decline of bicarbonate levels [34].

The survivor experienced metabolic alkalosis on DPE 8 as indicated by high levels of venous pH, bicarbonate, and base excess of the extracellular fluid (Figure 4c–e) [34]. This event initiated a compensated response to increase venous pCO_2_ levels near baseline post-DPE 8 (Figure 4a) [34]. No major differences of venous partial pressure of oxygen occurred throughout the time course, except increased levels for survivor and D13:non-survivor prior to euthanasia (pO_2_: Appendix A). Soluble oxygen percentages significantly decreased two-fold below baseline in all animals at DPE 8, and then percentages increased the following day (Appendix A).

In correlation to low blood pH, high lactate levels were observed in D10:non-survivor on DPE 10 and D13:non-survivor on DPE 13 (Appendix A). Hyponatremia was observed in all animals, with sodium levels less than 142 mmol/L, on DPE 10 (Appendix A) [32,33,35]. With chloride levels equal or less than 95 mmol/L, hypochloremia were initially observed in the survivor on DPE 9 as well as D13:non-survivor on DPE 12 (Appendix A). Low levels of sodium, chloride, and bicarbonate contributed to 17.9% decrease in anion gap in animals on DPE 10 (Appendix A and Figure 4d,f) [36]. No notable differences were observed in animals for ionized calcium, potassium, and glucose levels throughout the time course of aerosol LASV exposure (Appendix A). 

In comparison to baseline, hypoproteinemia occurred in all animals as total protein decreased 11% and albumin levels decreased 24.1% on DPE 8 (Figure 5a,b). All animals had alanine transaminase levels greater than maximum reference parameters (177 IU/L), except for D10:non-survivor (Figure 5c) [35]. Aspartate transaminase levels were greater than maximum reference parameters (135 IU/L) for all animals (Figure 5d) [35]. All LASV-exposed animals had alkaline phosphatase and gamma-glutamyl transferase levels above baseline, but levels were lower than maximum reference parameters (ALKP < 2633 IU, GGT < 136 IU/L, Figure 5e,f) [35].

Results of non-survivors did not exceed maximum parameters for serum blood-urea-nitrogen (36.7 mg/dL: Appendix A) nor serum creatinine (1.37 mg/dL: Appendix A) [35]. The survivor uniquely exceeded such parameters for creatinine initially on DPE 6 and blood-urea-nitrogen on DPE 13. Uric acid levels were non-uniformly increased in animals within DPE 8–13 in comparison to baseline (Appendix A). Total bilirubin levels did not change significantly throughout the time course (Appendix A). Overall, clinical chemistry identifies loss of ions and proteins in cardiovascular circulation when hypotension occurs during the decompensated stage.

### 3.5. Hematology

All LASV-infected animals experienced reduction in numbers of white blood cells and red blood cells that decreased hematocrit percentages on DPE 8 in comparison to DPE 0 (Figure 6a–c) [32,33,35]. In accordance with Adams et al. 2014, anemia was identified in all animals on DPE 8 with hemoglobin ≤ 10.4 g/dL and hematocrit percentage ≤ 34.8% (Figure 6c–d) [37]. White blood cell numbers decreased 61.9% on DPE 8 in comparison to DPE 0. Red blood cell numbers decreased 16.4% on DPE 8 in all animals but began to increase in survivor on DPE 30 (Figure 6b). Mean corpuscular volumes (MCV) and red blood cell distribution width percentages (%RDW) were within baseline levels throughout the time course [38]. The amounts of hemoglobin decreased 17.6% in blood of all animals on DPE 8 in contrast to DPE 0 (Figure 6d). During disease progression, non-survivors experienced no major change in mean corpuscular hemoglobin (MCH) and mean corpuscular hemoglobin concentrations (MCHC) [38] The survivor uniquely experienced high MCH (30.1 pg) and MCHC levels (43.1 g/dL) on DPE 5, then levels return near baseline on DPE 6. All animals had thrombocytopenia by DPE 8 as the numbers of platelets were less than lower limits (≤3.32 × 10^5^/µL; Figure 6e) [35]. On post-DPE 30, the survivor had elevated number of platelets.

Mean platelet volumes increase throughout the time course exceeding the highest reference levels (9.8 femtoliters) on DPE 8 whereas only the survivor had volumes that return to baseline on DPE 30 [33,35,38]. In comparison to DPE 0, prothrombin time shortened by 5.7% in all infected animals on DPE 2 and then increased 11.6% at DPE 13 (Figure 6f). The survivor had a reduction in prothrombin time on DPE 14, for which levels were sustained until euthanasia. International Normalized Ratios (INR) were between 0.9–1.2 ranges throughout the time course [38].

All animals had 31% increase in the number of neutrophils on DPE 4 followed by a 48% decrease at DPE 8, which associated in changes with number of white blood cells (Figure 6a,g) [32,35]. On DPE 13, the numbers of neutrophils return to baseline levels for survivor and D13:non-survivor. Monocytes numbers decrease 57.2% in NHPs on DPE 4 compared to baseline (Figure 6h). Lymphocyte numbers decrease 68.5% on DPE 4 and return near baseline levels on DPE 30 (Figure 6i). The percentages of neutrophils increase throughout time course for all animals with only the survivor returning to baseline on DPE 15 (Figure 6j). Low percentages of monocytes in blood occur on DPE 4–6, and lymphocytes on DPE 4–8 (Figure 6k,l).

### 3.6. LASV Immune Cells and Cytokine Levels in Blood

Flow cytometric analysis was performed to assess LASV glycoprotein-positive leukocytes during aerosolized challenge [31]. The percentage of CD4+ and CD8+ T cells as well as B cells positive for viral glycoprotein increased DPE 8–10 contrary to baseline (Appendix A). In correlation to the number of granulocytes in blood, the percentages of granulocytes positive for viral glycoprotein increased 53-fold on DPE 10 (Appendix A). Fold is defined by mean of all animals on DPE of interest divided by mean at DPE 0. The percentage of classical monocytes positive for viral glycoprotein increased 2.7-fold on DPE 7 (Appendix A) [31]. Two of four animals uniquely exhibited high percentage of classical monocytes positive for viral glycoprotein on DPE 2. Contrary to hematological numbers of monocytes, the percentages of activated monocytes positive for viral glycoprotein were 2.6-fold higher on DPE 4 when compared to baseline (Appendix A) [31]. Non-specific binding of antibodies was observed as LASV-positive classical and activated monocytes occurred on DPE 0.

Cytokine analysis of survivor and D13:non-survivor identified elevated plasma IFNγ levels on DPE 7–9 (Figure 7a) similar to reported human case [39]. Plasma IL-18, MCP1, and IL-6 levels were evident on DPE 11 (Figure 7b,c, Appendix A). On DPE 13, robust cytokine response occurred for IFNγ, MCP1, IL-18, IL1-RA, IL-1β, IL-2, IL-5, IL-6, IL-8, IL-12p40, MIP-1β, TNFα, GM-CSF, and G-CSF (Figure 7a–c, Appendix A). This robust cytokine response occurred relative to increased numbers of white blood cells (Figure 6a and Figure 7a–c).

### 3.7. Pathology

Pathological assessment was based on correlation of vital signs, clinical chemistry and related observations. Two of the four animals met euthanasia criteria on DPE 10, one on DPE 13, and one survived infection to be later euthanized at the study endpoint. Thereupon, a direct comparison between distribution and severity of lesions across time points cannot be made. Immediately following humane euthanasia, a complete necropsy was performed, and tissue collected for histopathology.

Immunoreactive alveolar histiocytes and pneumocytes corresponded with foci of necrosis in lung of D10:non-survivor (Figure 8(1a)). This non-survivor experienced alveolitis and alveolar edema with mild necrosis contrary to the normal alveolar architecture of survivor on DPE 41 (Figure 8(1b,1c)). Through in situ hybridization, genomic LASV RNA was detected in mononuclear cells and the lining of alveolar septae of D10:non-survivor in contrast to survivor on DPE 41 (Figure 8(1d,1e)).

At necropsy, the lungs of the non-survivors were mottled, failed to collapse completely and had clear, pinkish-red exudate in the thorax. Tracheal and submandibular lymph nodes were enlarged in all animals, except for one non-survivor. Inconsistent enlargement of mediastinal, bronchial, axillary, mesenteric, and inguinal lymph nodes were observed in non-survivors. The lungs of the survivor on DPE 41 were diffusely red, failed to collapse completely, and had multiple pleural fibrous adhesions spanning the left sided lung lobes and the diaphragm, pericardial sac, and thoracic wall. 

*Spleen-white pulp*: D10:non-survivors exhibit immunoreactive mononuclear cells within the germinal center of lymphoid follicles (Figure 8(2a)). Concurrent with lymphopenia, lymphocyte-depleted follicles were detected in a D10:non-survivor and contrast with the normal follicular appearance of the survivor on DPE 41 (Figure 8(2b,2c)). Genomic LASV RNA was detected in both the red and white pulp of DPE 10 non-survivors but was completely absent from the red and white pulp of the survivor on DPE 41 (Figure 8(2d,2e)).

*Spleen-red pulp:* D10:non-survivors exhibit strong and specific immunoreactive spindle-shaped cells, which are linearly arranged and likely represent endothelial and fibroblastic reticular cells (Figure 8(3a)). Varying degrees of red pulp mononuclear cell density and cellularity between non-survivors and the survivor on DPE 41 (Figure 8(3b,3c)). Genomic LASV RNA was detected within spindle-shaped cells of a D10:non-survivor but not the survivor on DPE 41 (Figure 8(3d,3e)).

*Liver*: D10:non-survivors exhibit intense hepatocellular immunoreactivity, which corresponds to foci of necrosis (Figure 8(4a)). Hepatocellular degeneration and necrosis with lipid-type vacuolar change in D10:non-survivor contrasts with the normal hepatocellular cord architecture of the survivor on DPE 41 (Figure 8(4b,4c)). LASV RNA was detected in clusters of hepatocytes within the enlarged liver of a non-survivor (Figure 8(4d)); however, it was not detected in the survivor on DPE 41 (Figure 8(4e)). 

*Adrenal gland*: D10:non-survivors exhibit strong multifocal adrenal cortical cell membrane and cytoplasmic associated immunoreactivity (Figure 8(5a)). Degeneration and necrosis of adrenal cortical cells with congestion of the cortical interstitium contrasts with the normal architecture of survivor on DPE 41 (Figure 8(5b,5c)). Viral RNA was detected in the cortical cells of a D10:non-survivor, but not in the survivor on DPE 41 (Figure 8(5d,5e)). 

*Heart*: D10:non-survivors exhibit immunoreactivity limited to mononuclear and spindle-shaped cells within the interstitium (Figure 8(6a)) as well as multifocal histiocytic and lymphoplasmacytic inflammation (Figure 8(6b)). Note the wide cuff of mononuclear inflammation surrounding a blood vessel within the myocardium of the survivor on DPE 41 (Figure 8(6c)). Genomic LASV RNA was detected within inflammatory foci of the D10:non-survivor (Figure 8(6d)); however, none was detected in the survivor on DPE 41 (Figure 8(6e)).

*Kidney*: D10:non-survivors exhibit immunoreactive tubule epithelial cells (Appendix A) as well as multifocal degeneration and necrosis with minimal interstitial mononuclear inflammation (Appendix A). Perivascular mononuclear inflammation extends into the interstitium of the survivor on DPE 41 (Appendix A). Genomic LASV RNA was detected in the inflamed interstitium of a D10:non-survivor (Appendix A), but not the survivor on DPE 41 (Appendix A).

*Urinary bladder*: D10:non-survivors exhibit immunoreactive transitional epithelial cells (Appendix A). This non-survivor experienced degeneration and necrosis of transitional epithelial cells (Appendix A). Normal transitional epithelium and the presence of rare submucosal mononuclear cells were evident in the survivor on DPE 41 (Appendix A). Genomic LASV RNA was identified in the transitional epithelium of a D10:non-survivor; however, not in the survivor on DPE 41 (Appendix A).

*Tonsil:* D10:non-survivors exhibit immunoreactive tonsillar epithelium (Appendix A). Multifocal tonsillar epithelial cell degeneration and necrosis is observed in a D10:non-survivor; however, normal lymphoid follicular structure occurs in the survivor on DPE 41 (Appendix A). Genomic LASV RNA was detected in the stratified squamous epithelium and lymph nodules in the tonsil of a D10:non-survivor, but not in the tonsil of the survivor on DPE 41 (Appendix A).

*Cerebellum*: D10:non-survivors exhibit immunoreactive endothelial and scattered single cells (Appendix A). This animal exhibits mononuclear perivascular inflammation (Appendix A). Likewise, the survivor on DPE 41 also exhibits a mononuclear perivascular inflammation but the tissue is more organized and less disruptive to the adjacent neuropil (Appendix A). Genomic LASV RNA was detected in the vessels of the cerebellum of a D10:non-survivor, but not in the vessels of the survivor on DPE 41 (Appendix A).

*Pituitary gland*: D10:non-survivors exhibit immunoreactive cells that correlate to foci of increased cellularity and blood vessels in the pituitary gland (Appendix A). The survivor on DPE 41 exhibits normal cellularity of the pituitary gland on DPE 41 (Appendix A). Genomic LASV RNA was detected in the pituitary gland of D10:non-survivor, but not in the pituitary gland of the survivor on DPE 41 (Appendix A).

All animals had adequate subcutaneous, omental, abdominal, and retroperitoneal adipose tissue. Lassa virus endothelial cell immunoreactivity is present in nearly every non-survivor tissue section. In addition, all non-survivors exhibited mild multifocal meningitis and encephalitis as well as foci of inflammation and necrosis within the epithelium of the gastrointestinal tract [38]. The survivor’s liver, heart, kidney, pancreas, brain, and large intestine, on day 41 post-exposure, contained vessels with evidence of vasculitis or ongoing vascular damage. The lymphoid follicles within the lymph nodes and spleen were frequently hyperplastic in survivor [38]. No detectable LASV were observed in plasma titer (Figure 3) or ISH of any tissue sections examined of survivor at DPE 41 (Figure 8, Appendix A). 

## 4. Discussion

Lassa virus has a high mortality rate and is easily transmitted through either direct contact with the rodent itself or contact/inhalation of its excreta [5,6,11]. Reports of aerosol transmission of Lassa virus as well as direct contact with infected individuals or bodily fluids have been identified in metropolitan areas [6,10,11,13,14,15]. Okokhere et al. identified 291 patients treated for Lassa fever in Nigeria between 2011 and 2015 [2]. These patients experience fever, respiratory distress, and fluid loss associated with diarrhea, vomiting, and bleeding. Acute kidney injury was observed in 28% of patients and central nervous system complications occurred in 37% of cases [2]. In view of the limited extent of experimental history in humans, this natural history study of acute aerosol exposure of non-human primates identified disease progression within a controlled environment to enable hypothesis generation for human Lassa infection. In addition, this study adds a state-of-the-art road map of an animal model to complement future clinical trials of novel vaccines and therapeutics [40]. A well-characterized animal model will also be useful where bridging studies are required to explore impacts on hearing loss, infection models of pregnant animals, or immunogenicity complexities of Lassa virus [19,41].

Our findings identify viremia primarily at the beginning of the clinical stage and peak levels occur within the decompensated stage on day 10–13. Evidence of the virus in blood and all examined tissues are found in all non-survivors. Virus was also found in blood of survivor, but not detectable 41 days after infection. In addition, Lassa virus was not detected in target tissues of the survivor such as liver, spleen, and adrenal gland nor the urinary bladder.

All non-human primates encountered increased temperatures and heart rates within the clinical stage. Similar to a severe human case [42], the survivor experiences bimodal temperature that later return to baseline levels. These changes in the survivor correlate with transient increase in blood cells and cytokines followed by a lack of detectable Lassa virus in blood and tissues. Commonly observed in viral hemorrhagic fevers, the elevated heart rate has been linked to cell signaling events for vascular endothelial dysfunction which likely explains the transient increase in heart rate and temperature in the survivor [43,44]. Several organs of the survivor, such as the heart, kidney, liver, pancreas, brain, and large intestine, bear vasculitis lesions at day 41 in comparison to intramuscular studies of surviving NHPs beyond the acute stage [19].

Of note, all NHPs had electrocardiographic findings during the course of disease beyond the expected sinus tachycardia ranging from PR depressions, ST/T segment changes to decreased QRS voltage. Inflammation of the heart was predominantly histiocytic and lymphoplasmacytic as well as observations of random foci of cardiac myocyte degeneration were scattered throughout the interstitium. These observations merit additional examination in future studies and should include evaluation of cardiac enzymes during the course of disease. In comparison, humans can have nonspecific ST and T wave changes on electrocardiogram, myocarditis, and more rarely pericardial effusions [45,46,47,48].

Immune dysregulation occurs early in the clinical stage of aerosol induced Lassa fever. Migration of blood components into surrounding tissues is associated with fever and viremia as well as decreased anion gap, albumin, and lymphocytes levels. Though monocytes levels decrease on the initial day of clinical stage, an increase trend of activated monocytes positive for viral glycoprotein were observed above baseline levels. Several studies report the nucleoprotein of LASV inhibits retinoic acid-inducible gene 1 (RIG-I) antiviral responses, and inhibits antigen presenting cell-lymphocyte responses [49,50,51]. In light of this dysregulation, blood cytokine levels are low early in the clinical stage of disease. During the decompensated stage, blood cytokine levels increase in parallel with leukocyte numbers. All non-survivors experienced splenomegaly, hepatomegaly, and lymphadenopathy. These animals present inconsistently enlarged mediastinal, bronchial, auxiliary, mesenteric, and inguinal lymph nodes. The survivor bears mildly diffuse follicular hyperplasia in spleen and all respective lymph nodes.

Acute kidney injury is significantly associated with death in human cases of Lassa fever [2,6,52,53]. In present study, changes in renal function in non-survivors were initially identified by the decline in venous partial pressure of carbon dioxide that shifts chemical equilibria towards alkalosis. Loss of cations, anions, and protein in cardiovascular circulation during the clinical and decompensated stage decreased the anion gap. Upregulation of renal function markers, blood urea nitrogen and creatinine, occurred late in the decompensated stage. Two of three non-survivors experienced metabolic acidosis a day prior to euthanasia. The survivor has renal function markers that return to baseline levels during recovery. Histological assessment of the survivor further identified renal perivascular inflammation as we previously described [19].

Though it is unclear the reason for the reduction of disease severity, the survivor maintained hemoglobin levels and dietary intake during the clinical stage of disease. Evidence of hemoglobin synthesis in survivor, in association with the Haldane effect, increased total carbon dioxide and bicarbonate levels contrary to observations of non-survivors. Metabolic alkalosis was observed in the survivor contrary to respiratory alkalosis in non-survivors. It appears high baseline hemoglobin levels in survivor may have improved the outcome of the disease. Based on these findings, acute kidney injury and migration of blood components during Lassa disease affect blood pH regulation.

Renal dysfunction is common in severe cases of viral hemorrhagic fevers such yellow fever and dengue virus [54]. Unlike acidosis observed early in sepsis, alkalosis has been observed in clinical stage of severe Ebola and Marburg virus infections [55,56,57,58]. Belonging to *Arenaviridae* family with Lassa virus, Machupo virus has also been linked to respiratory alkalosis based on a severe case of Bolivian hemorrhagic fever [59]. Thereupon, changes in blood pH levels could be used as indicators of disease severity during viral hemorrhagic fevers.

Clinical and laboratory predictors of severe Lassa virus infection respectfully include fever, tachycardia, lymphocytopenia, changes in anion gap and venous partial pressure of carbon dioxide followed by alkalosis, anemia, and hypoproteinemia. Immediate treatment is needed to prevent hypovolemic shock and death. Use of clinical and laboratory predictors include management of early antiviral therapy, fever, ventilation, intravenous volume with supplementation, nutrition, blood transfusion when anemic, and hemodialysis may improve chances of survival of infected patients.

## 5. Conclusions

This natural history study, performed with telemetric analysis, has provided detailed clinical, laboratory, and pathologic parameters for aerosol induced Lassa virus infection in NHPs. This study revealed many similarities between NHP and human Lassa disease, which suggests that this may be a reasonable model for medical countermeasure development for both endemic disease and traditional biodefense purposes.

## Figures and Tables

**Figure 1 viruses-12-00593-f001:**
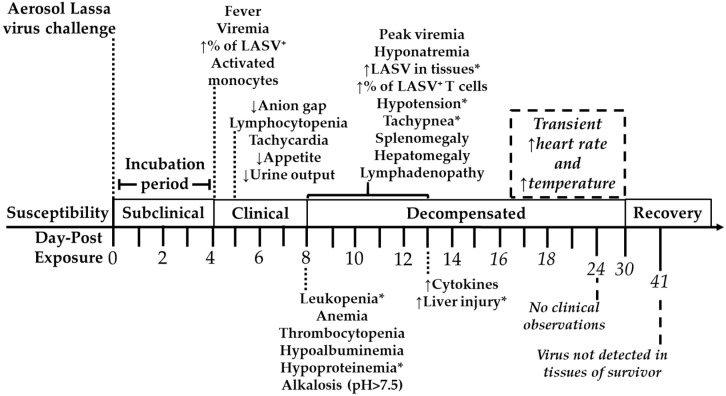
Natural history of aerosol induced Lassa virus infection of cynomolgus macaques. Subclinical stage of disease progression is marked by mean clinical score of zero from the day of challenge until presentation of fever and virus (day-post exposure, DPE 0–3). Clinical stage of infection involved detection of fever and virus (DPE 4–7). Further trend of viral glycoprotein-positive activated monocytes on DPE 4 as well as low anion gap, tachycardia, lymphocytopenia, decrease appetite and urine output on DPE 5. Decompensated stage (DPE 8–30) was identified with mean clinical score ≥0.4, peak viremia, alkalosis with blood pH > 7.5, and elevated serum levels of liver and kidney function markers. Leukopenia, anemia, thrombocytopenia, hypoalbuminemia, and hypoproteinemia were observed by DPE 8. Soluble oxygen percentage and mean arterial pressure are substantially decreased in comparison to prior time points. Peak viremia was followed by mild IFNγ response. On DPE 12–13, robust cytokine responses were associated with significant increased number of white blood cells, biochemical liver injury markers, and blood urea nitrogen levels. The decompensated stage included random display of posture and appearance as well as impairment of neurological function, hepatomegaly, splenomegaly, and lymphadenopathy. The survivor experienced transient increase in heart rate and temperature within DPE 25–29. Recovery stage (Post-DPE 30) was establishing based on mean clinical score of zero for eleven days, viremia below 100 PFU/mL, blood pH < 7.5, and liver and kidney functions markers near baseline levels. * Asterisks marks exception of one animal.

**Figure 2 viruses-12-00593-f002:**
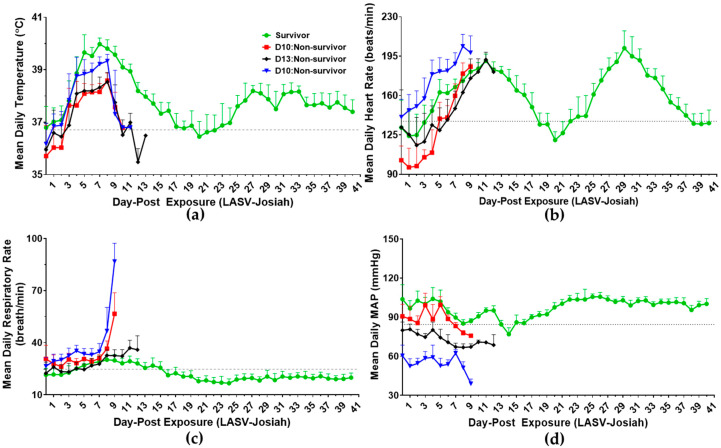
Mean daily telemetric assessment of aerosol Lassa virus infected cynomolgus macaques. Mean daily (**a**) temperature, (**b**) heart rate, (**c**) respiratory rate, and (**d**) mean arterial pressure (MAP). (● Survivor) (■ D10:Non-survivor) (♦ D13:Non-survivor) (▼ D10:Non-survivor). Grey dashed line is average relative to each animal for DPE 5 to DPE 0. Error bars indicate standard deviation.

**Figure 3 viruses-12-00593-f003:**
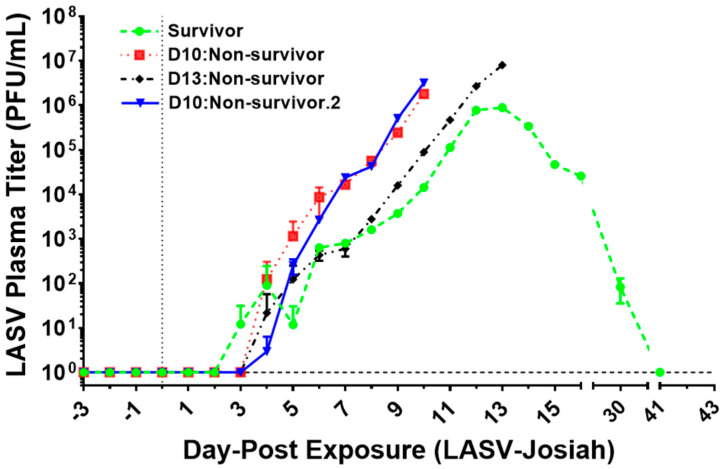
Time course of Lassa virus detection in plasma of challenged cynomolgus macaques. Detection was conducted using qRT-PCR to amplify RNA using specific primers for the S segment of the Lassa virus (LASV) glycoprotein gene. DPE 0 marked by vertical dot line and average by horizontal dash line. (● Survivor) (■ D10:Non-survivor) (♦ D13:Non-survivor) (▼ D10:Non-survivor).

**Figure 4 viruses-12-00593-f004:**
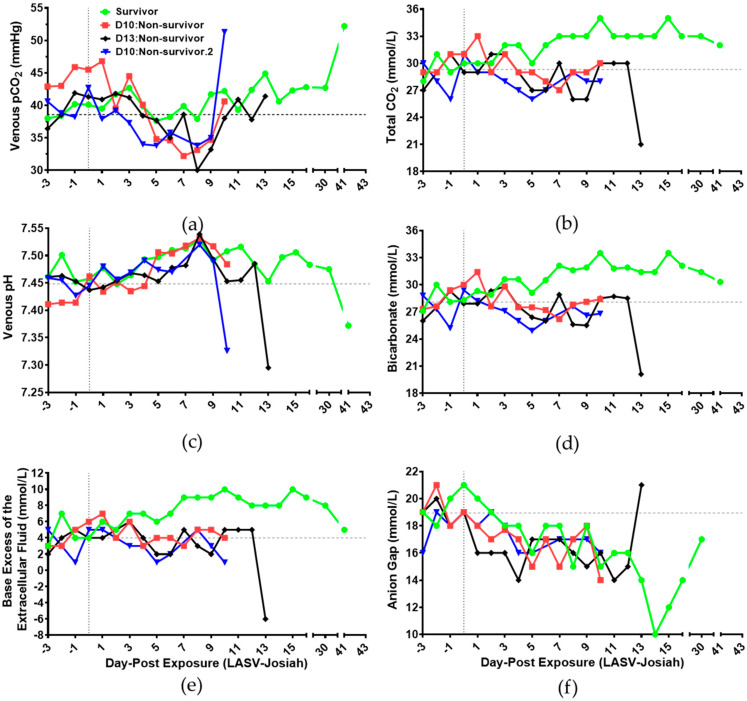
Time course of blood gases and related factors of Lassa virus-challenged cynomolgus macaques. Detection of (**a**) partial pressure of carbon dioxide: pCO_2_, (**b**) total CO_2_, (**c**) pH, (**d**) bicarbonate, and (**e**) base excess of extracellular fluid was determined in whole blood using i-Stat CG4+. Assessment of (**f**) anion gap in plasma involved iStat-1 CHEM8+. (● Survivor) (■ D10:Non-survivor) (♦ D13:Non-survivor) (▼ D10:Non-survivor.2) (Vertical grey dotted line: DPE 0) Grey dashed line is average relative to each animal for DPE 3 to DPE 0.

**Figure 5 viruses-12-00593-f005:**
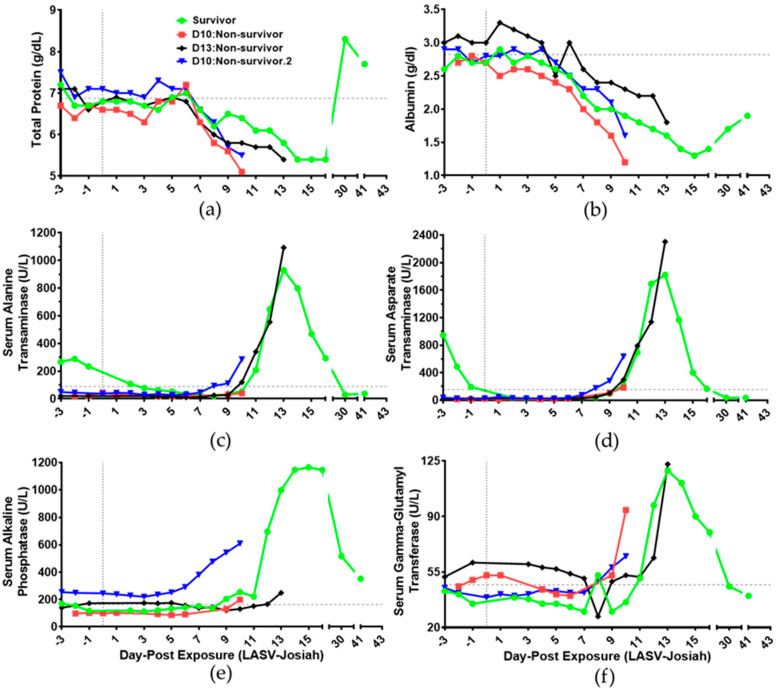
Time course of plasma proteins and liver function enzymes of Lassa virus-challenged cynomolgus macaques. Detection of (**a**) total protein and (**b**) albumin in plasma of whole blood using Piccolo chemistry. Serum levels of (**c**) alanine transaminase, (**d**) aspartate transaminase, (**e**) alkaline phosphatase, and (**f**) gamma-glutamyl transferase were detected in serum using Piccolo chemistry. (● Survivor) (■ D10:Non-survivor) (♦ D13:Non-survivor) (▼ D10:Non-survivor.2) (Vertical grey dotted line: DPE 0) Grey dashed line is average relative to each animal for DPE 3 to DPE 0.

**Figure 6 viruses-12-00593-f006:**
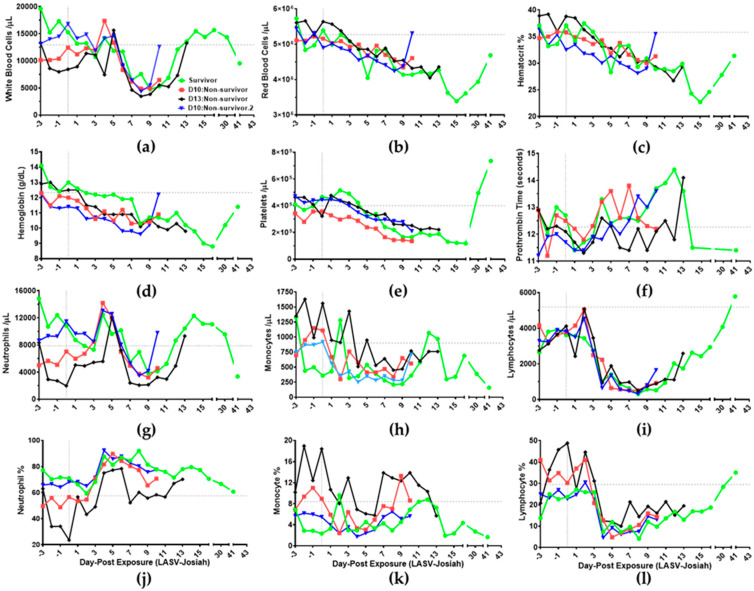
Time course analysis of blood cells and circulatory components from aerosol LASV-infected cynomolgus macaques. (**a**) Number of white blood cells, (**b**) number of red blood cells, (**c**) hematocrit percentage, (**d**) amount of hemoglobin (**e**) number of platelets, (**f**) prothrombin time, (**g**) number of neutrophils, (**h**) number of monocytes (**i**) number of lymphocytes, (**j**) percentage of neutrophils, (**k**) percentage of monocytes, and (**l**) percentage of lymphocytes detected per microliter of blood by Hemavet 950FS hematology analyzer. Prothrombin time (**f**) was detected using iStat1 PT/INR. (● Survivor) (■ D10:Non-survivor) (♦ D13:Non-survivor) (▼ D10:Non-survivor.2) (Vertical grey dotted line: DPE 0) Grey dashed line is average relative to each animal for DPE 3 to DPE 0.

**Figure 7 viruses-12-00593-f007:**
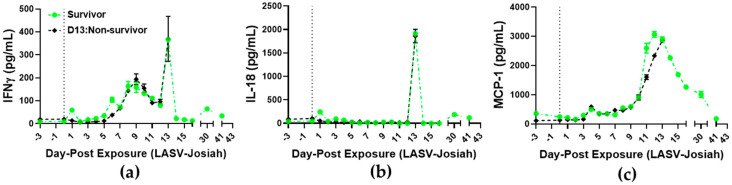
Analysis of immune responses in blood from aerosol LASV-infected cynomolgus macaques. Cytokine analyses of blood, (**a**) IFNγ, (**b**) IL-18, and (**c**) MCP-1, were performed on samples collected 3 days prior to exposure and DPE 0–41. (● Survivor) (♦ D13:Non-survivor).

**Figure 8 viruses-12-00593-f008:**
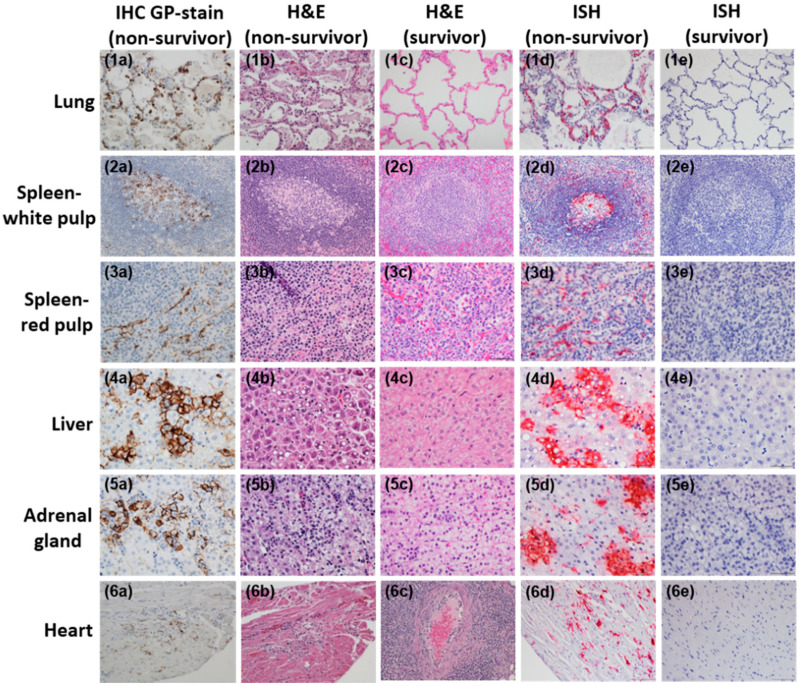
Pathological assessment of tissues of aerosol infected non-human primates. Lung (**1a**–**1e**, 20×), spleen-white pulp (**2a**–**2e**, 40×), spleen-red pulp (**3a**–**3e**, 40×), liver (**4a**–**4e**, 40×), adrenal gland (**5a**–**5e**, 40×), and heart (**6a**–**6e**, 20×). Tissues were from ● survivor on DPE 41 (all tissues), ■ D10:non-survivor (lung and heart) and ▼ D10:non-survivor.2 (liver, spleen tissues, and adrenal gland). Tissue sections embedded in paraffin were stained for LASV GP2 in non-survivor (IHC-GP: **1a**–**6a**), H&E stain for non-survivor on DPE 10 (**1b**–**6b**), and H&E stain for survivor on DPE 41 (**1c**–**6c**). Detection of viral genomic RNA was performed using nucleic acid probe, targeting the polymerase gene with the L segment at sequence 466–1433, in non-survivors on DPE 10 (**1d**–**6d**) and absent in the survivor on DPE 41 (**1e**–**6e**) on far right column.

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
