# Peer review of "Natural History of Aerosol Induced Lassa Fever in Non-Human Primates"

_viruses, 2020, doi:10.3390/v12060593_

Round 1

Reviewer 1 Report

General comments:

The manuscript by Downs and colleagues describes a study aimed at providing a comprehensive assessment of the clinical features of cynomolgus macaques exposed to aerosolized Lassa virus (LASV), the causative agent of Lassa fever (LF), which poses a major public health concern in LF endemic West African countries. LASV causes chronic infections in a natural reservoir rodent, Mastomys natalensis, and infected rodents shed infectious virions in their excreta. Humans are infected with LASV through mucosal exposure to aerosols or direct contact of abraded skin with fomites. Despite substantial and continuous efforts since the discovery of LASV 50 years ago, there are no FDA-approved vaccines or treatments available for LF. Non-human primate (NHP) models for LASV infections mimic human LF cases more closely than small animal models using mice and guinea pigs, and are important for the development of medical countermeasures. Infections with LASV in NHP models thus far have been mainly induced by intramuscular (i.m.) or subcutaneous (s.c.) inoculation. There have been few reports of aerosolized LASV-infected NHP studies and the disease course has been incompletely assessed. To address this issue, the authors performed aerosolized LASV infection to NHPs and monitored various disease parameters specific to human LF cases in detail. NHPs infected with aerosolized LASV exhibited LF symptoms typical in humans, including fever, subdued behavior, facial edema, and petechiae. A state-of-the-art, less invasive monitoring system revealed viremia, elevated hepatic enzymes, pancytopenia, abnormalities in ECG, and liver lesions, also commonly observed in human FL cases.

This paper is well written and the experimental section is carried out well. The results are for the most part clearly presented. The cause of disease observed in this study reflects that of human LF cases and is similar to that seen in other NHP models using i.m. and s.c. routes. However, the advantage of using the aerosolized infection system, which requires special devices, over i.m. or s.c. inoculations is not evident. Despite this limitation, the information presented in this manuscript represents an advance in our knowledge about the disease course of NHPs infected with LASV aerosolized LASV, and will support future basic and translational research on aerosol-induced LASV infection, an important natural infection route of LF. There are some aspects of this work that could be improved by providing additional clarifications as pointed below.

Specific comments:

  1. Line 86: The size of cynomolgus macaques used in this study have relatively high variations (4 kg to 9 kg). Is there any correlation between the size of animal and outcome of the disease?
  2. The disease course is nicely summarized in Fig. 1. It would be also informative to show the transition of disease severity (e.g. healthy, mild, moderate, severe, and dead) or clinical scores for individual animals to show the variation in disease course.
  3. Line 236: Sensorineural hearing loss is an important long-term sequela of Lassa Fever. Did the authors conduct subjective or objective measurements for healing functions? Related to this issue, it would be informative to provide pathology data of inner ear sections if the authors performed it.
  4. Line 248: “Based on combine observation of posture, appearance…” This conclusion is not supported by any data. The authors need to provide information on the clinical scores of each animal.
  5. Lines 274-277: The ECG results will be hard to understand by non-specialists. It would be helpful to incorporate interpretations of ECG findings, such as PVCs, PR depressions, and decreased QRS complex voltage, in this animal model.
  6. Line 320 and other parts of this manuscript: It is confusing that two animals euthanized on 10 day-post exposure (DPE) have the same name (D10:Non-survivor). The authors should give different names or IDs to individual animals.
  7. Figures 4, 5, 6, 7, S6, S7, S8, and S9: The X-axis should be split as the authors did in Fig. 3.
  8. Section 3.7. Pathology: The writing in this section might be misleading and the authors need to carefully describe the results. The authors cannot simply blame the differences in pathological findings on the differences between non-survivor and survivor. The tissue sections from non-survivors were collected on 10 DPE, when the virus was actively multiplying, and tissue sections from the survivor were collected on 41 DPE, when the virus was already cleared. In addition, there are no data for the survivor at 10 DPE.
  9. Lines 431-432 (similar concern raised in comment #8): The survivor at 10 DPE might have had an empty stomach. The authors need to clearly present the results. For example, “The survivor at 41 DPE did not have an empty stomach, …”.
  10. Fig. 8: It would be informative to incorporate the IHC data on the survivor at 41 DPE. This might also support the authors’ argument described at line 491.

Minor points:

Line 49: “Lassa’s” should be “Lassa Fever”.

Lines 632-634: Reference 14 lacks page number information.

Author Response

  First and foremost, the authors of this manuscript greatly appreciate the significant contribution made to review this article and provide feedback to make improvements.  Thank you very much.  Below, are responses to comments (REVIEWER 1 Comments in italics).

    Line 86: The size of cynomolgus macaques used in this study have relatively high variations (4 kg to 9 kg). Is there any correlation between the size of animal and outcome of the disease?

RESPONSE: The weight of males used in the initial studies were between 8.2-8.9 kg. L92 in manuscript is corrected to remove high variation. L249 was changed to “This challenge dose was fatal for three of four exposed macaques with body weight loss observed in D10:non-survivor (1.9 kg), D10:non-survivor.2 (1.3 kg), and D13:non-survivor (3.9 kg).” L256 was changed to “No signs of illness were identified in post-DPE 30 assessments in single survivor with body weight loss (2.5 kg) when euthanized on DPE 41 (Table S1).”  Though both D10:Non-survivor and D10:Non-survivor.2 initially weighed 8.9 kg, this weight was not drastically different from Survivor (8.8 kg) and D13:Non-survivor (8.2 kg). No correlation can be made with the limited number of animals based on animal weight and outcome of disease.

    The disease course is nicely summarized in Fig. 1. It would be also informative to show the transition of disease severity (e.g. healthy, mild, moderate, severe, and dead) or clinical scores for individual animals to show the variation in disease course.

RESPONSE: Per request, the disease course summarized in Fig. 1 is associated clinical parameters, laboratory parameters, pathology and clinical assessment scores of non-human primates during the natural history study seen in supplementary material (Table S1).  Clinical assessment scores are derived from detailed descriptions in the clinical assessment in accordance to institutional guidelines. Score codes during clinical assessment are normal (0), very mild (1), mild (2), moderate (3), and severe (4) for responsiveness/behavior, rash, bleeding, gastrointestinal (GI) symptoms, posture/appearance, and respiration. Score codes during the assessment are normal (0), mild (1), moderate (2), and severe (3) for food consumption, urine output, and edema.  Vocalization assessment includes normal (0) and in distress (1).  Exudate assessment includes normal (0) and present (1). A mean clinical score for all animals have been recorded for each day-post exposure (DPE).  This methodology is described in L105 in section 2.3. Daily Observations.

    Line 236: Sensorineural hearing loss is an important long-term sequela of Lassa Fever. Did the authors conduct subjective or objective measurements for healing functions? Related to this issue, it would be informative to provide pathology data of inner ear sections if the authors performed it.

RESPONSE:  This experimental research was performed in 2010. No formal hearing studies were conducted as assessments and inner ear sampling of animals started in 2011. Clinical assessments score of NHPs have been added in the manuscripts supplementary (Table S1) to account for responsiveness/behavior, posture/appearance, neurological function, vocalization and other clinical signs (rash, bleeding, gastrointestinal (GI) symptoms, urine output, edema, respiration, and exudate).  A mean clinical score for all animals have been recorded for each day-post exposure (DPE).  

    Line 248: “Based on combine observation of posture, appearance…” This conclusion is not supported by any data. The authors need to provide information on the clinical scores of each animal.

RESPONSE: L248, now L254-L256, has been modified in accordance with Table S1-described in previous comment. This line has the following statement:  “The highest mean clinical score of the survivor occurred on DPE 13 (Table S1). The survivor’s average assessment score of posture and appearance as well as neurological function score was mild (2) compared to moderate endpoint scores for D10:non-survivor (2.5), D13:non-survivor (2.5), and D10:non-survivor.2 (3.5).”

    Lines 274-277: The ECG results will be hard to understand by non-specialists. It would be helpful to incorporate interpretations of ECG findings, such as PVCs, PR depressions, and decreased QRS complex voltage, in this animal model.

RESPONSE: ECG results (Fig. S3-S5) have been marked to help non-specialist understand data for PVCs, PR depressions, and decreased QRS complex voltage.

    Line 320 and other parts of this manuscript: It is confusing that two animals euthanized on 10 day-post exposure (DPE) have the same name (D10:Non-survivor). The authors should give different names or IDs to individual animals.

RESPONSE: The two D10:non-survivors have been designated as D10:Non-survivor and D10:Non-survivor.2 as recommended.  The choice of names were to distinguish the survivor versus non-survivors, and day the non-survivors died.

    Figures 4, 5, 6, 7, S6, S7, S8, and S9: The X-axis should be split as the authors did in Fig. 3.

RESPONSE: Corrections made to the main figures and supplementary material.

    Section 3.7. Pathology: The writing in this section might be misleading and the authors need to carefully describe the results. The authors cannot simply blame the differences in pathological findings on the differences between non-survivor and survivor. The tissue sections from non-survivors were collected on 10 DPE, when the virus was actively multiplying, and tissue sections from the survivor were collected on 41 DPE, when the virus was already cleared. In addition, there are no data for the survivor at 10 DPE.

RESPONSE: A noteworthy point is made and the following corrections are added at the beginning of 3.7 Pathology at L418: “Two of the four animals met euthanasia criteria on DPE 10, one on DPE 13, and one survived infection to be later euthanized at the study endpoint. Thereupon, a direct comparison between distribution and severity of lesions across time points can not be made. Immediately following humane euthanasia, a complete necropsy was performed and tissue collected for histopathology.”

    Lines 431-432 (similar concern raised in comment #8): The survivor at 10 DPE might have had an empty stomach. The authors need to clearly present the results. For example, “The survivor at 41 DPE did not have an empty stomach, …”.

RESPONSE: The lines 431-432 in the Pathology section have been removed.

    Fig. 8: It would be informative to incorporate the IHC data on the survivor at 41 DPE. This might also support the authors’ argument described at line 491.

RESPONSE: Genomic LASV RNA was assessed through in situ hybridization (ISH) in both non-survivor at DPE 10 and survivor at DPE 41. Though informative, IHC of survivor on DPE 41 was precluded relative to limitations. Respectfully, use of ISH and LASV plasma titer provide the molecular pathological assessment of virus in tissue and detection within plasma of non-survivor at DPE 10. In comparison D10:Non-survivor, the survivor had high plasma titer early in infection but no major indications of the virus itself later as described in L502-L504: “No detectable LASV were observed in plasma titer (Fig. 3) or ISH of any tissue sections examined of survivor at DPE 41 (Fig. 8, Fig. S10).”

Minor points:

Line 49: “Lassa’s” should be “Lassa Fever”.

RESPONSE: Change has been made as recommended.

Lines 632-634: Reference 14 lacks page number information.

RESPONSE: Corrections are implemented to the related references.

Reviewer 2 Report

The authors present a Lassa fever animal model with important experimental data sustaining the adequacy of the model for future vaccine or therapy development. Most importantly the virus stability and aerosol infectivity are detailed.

TITLE

Line 2: Title is not appropriate, it is about developing an animal model, the is not about "Natural History"

ABSTRACT

L26 : These numbers cases/death need to be updated with the recent publications (i.e. 10,000 death yearly...)

L26 "deliberate aerosol" classes Lassa virus in the select agent list. It is also a CAT A list under different criteria (e.g. rarely seen in the United States. risk to national security easily disseminated or transmitted from person to person; high mortality rates, public health ...

L28 Vaccine is the problem and need a animal model

L31 Largely analogous to several (not all) symptoms including...

INTRODUCTION

L44 as for L26 Update the numbers

L46 ref to L26 (different criteria?) CA or Select Agent or both.

L48-49: Diagnostic available in Nigeria and Sierra Leone and Liberia

L47- L53 redundancy

L61 Lassa virus Josiah strain (ICTV)

L68 "establish the natural history" ? Show the evolution of LF in animal model using the aerosol transmission which appear one of the major transmission mode in the natural environment... 

L74 LASV Josiah strain

L77 with the head only....

L105 were humanly euthanized

L222 separate the figure title form the comment (legend ? caption ? or text for results? "Pre-clinical phase ....."

L237 Authors have to definitely explain "the natural history of ..."?

L249 Any specific neuronal symptom ? This is classical in Lassa fever recovery (e.g. deafness).

Fig 2 : Heart rate of the survivor bimodal ?

L343 : Survivor: remarquable pic alkaline phosphatase and gamma-glutamyl transferase - explain? Fig 5

L414 Fig 8: Remarkable Pathological assessment

L500 "experimental history"

L500 also "limited extend" not for human and Cynomolgus model has been studied before...

L519 and 561 see also " High heart rate at admission as a predictive factor of mortality in hospitalized patients with Lassa fever: An observational cohort study in Sierra Leone. J Infect. 2020 Feb 3. pii: S0163-4453(20)30061"

L548 Important discussion "why one survived?" see L 343

Two important point to present/discuss

- At first: The emerging geographic extension of the Lassa fever in West Africa (as well as exported cases) strongly favor of this type of research on LASV

- L 505/ Neural system (see  Animal Model of Sensorineural Hearing Loss Associated with Lassa Virus Infection. J Virol. 2015 Dec 30;90(6):2920-7. doi: 10.1128/JVI.02948-15.

Author Response

  First and foremost, the authors of this manuscript greatly appreciate the significant contribution made to review this article and provide feedback to make improvements.  Thank you very much.  Below, are responses to comments (REVIEWER 2 Comments in italics).

TITLE

Line 2: Title is not appropriate, it is about developing an animal model, the is not about "Natural History”

RESPONSE: The concern of the term “Natural History” is appropriate and fosters a detailed explanation. According to the FDA (https://www.fda.gov/regulatory-information/search-fda-guidance-documents/product-development-under-animal-rule), “Natural history studies are studies in which animals are exposed to a challenge agent and monitored to gain an understanding of the development and progression of the resulting disease or condition, including parameters such as manifestations (e.g., signs, clinical and pathological features, laboratory parameters, extent of organ involvement, morbidity, and outcome), the time from exposure to manifestation onset, time course and order of manifestation progression, and severity.”  This detailed aerosol study collected health information regarding clinical observations, state of the art vital sign monitoring, continuous ECG surveillance, daily virus detection, and end-point pathological assessment.  Daily analysis of blood gas, clinical chemistry, cytokines/chemokines and cytometric assessment of immune cells extensively expands health information for disease progression.

Per request, the disease course summarized in Fig. 1 is associated clinical parameters, laboratory parameters, pathology and clinical assessment scores of non-human primates during the natural history study seen in supplementary material (Table S1).  Clinical assessment scores are derived from detailed descriptions in the clinical assessment in accordance to institutional guidelines. Score codes during clinical assessment are normal (0), very mild (1), mild (2), moderate (3), and severe (4) for responsiveness/behavior, rash, bleeding, gastrointestinal (GI) symptoms, posture/appearance, and respiration. Score codes during the assessment are normal (0), mild (1), moderate (2), and severe (3) for food consumption, urine output, and edema.  Vocalization assessment includes normal (0) and in distress (1).  Exudate assessment includes normal (0) and present (1). A mean clinical score for all animals have been recorded for each day-post exposure (DPE).  

ABSTRACT

L26 : These numbers cases/death need to be updated with the recent publications (i.e. 10,000 death yearly...)

RESPONSE: The recommended changes have been made.

L26 "deliberate aerosol" classes Lassa virus in the select agent list. It is also a CAT A list under different criteria (e.g. rarely seen in the United States. risk to national security easily disseminated or transmitted from person to person; high mortality rates, public health ...

RESPONSE: Adjustments are made relative to the cited material from the CDC and Federal Select Agent Program.

L28 Vaccine is the problem and need a animal model

RESPONSE: Content related to vaccine and need for an animal model have been included in L28-L29.

L31 Largely analogous to several (not all) symptoms including...

RESPONSE: The direction provided is greatly appreciated. By medical definition, symptoms is any indication of disease perceived by the patient.  L510-L511 describes what humans experience. In the abstract, L31 is intended to make a comparison of signs observed in the animal model and human disease. Corrections have been made. 

INTRODUCTION

L44 as for L26 Update the numbers

RESPONSE: Changes have been made according to recommendations.

L46 ref to L26 (different criteria?) CA or Select Agent or both.

RESPONSE: Modifications have been implemented, with source citations, indicating LASV is both a CDC category A biological agent and on the select agents and toxins list.

L48-49: Diagnostic available in Nigeria and Sierra Leone and Liberia

RESPONSE: The reviewer comment is noted and the sentence in L48-L49 has been removed.

L47- L53 redundancy

RESPONSE: L52-L53 have been removed to reduce appearance of redundancy.

L61 Lassa virus Josiah strain (ICTV)

RESPONSE: The text has been added “Lassa virus Josiah strain (NCBI:txid11622).

L68 "establish the natural history" ? Show the evolution of LF in animal model using the aerosol transmission which appear one of the major transmission mode in the natural environment...

RESPONSE: Similar to initial response, the FDA Animal Rule Guidance definition with citation has been added to L70-L73.

L74 LASV Josiah strain

RESPONSE: The change has made.

L77 with the head only....

RESPONSE: The text has been added to the manuscript.

L105 were humanly euthanized

RESPONSE: This phrase has been added.

L222 separate the figure title form the comment (legend ? caption ? or text for results? "Pre-clinical phase ....."

RESPONSE: Figure 1 is used to identify stages of disease progression and observations relative to broadly spread-out results. When first presenting the stage of disease progression, the name of the stage will be the first word presented in the caption. The figure legend title is bolded and all other figures throughout the manuscript.

L237 Authors have to definitely explain "the natural history of ..."?

RESPONSE: The FDA Animal Rule Guidance definition with citation has been added to L70 as previously discussed above.

L249 Any specific neuronal symptom ? This is classical in Lassa fever recovery (e.g. deafness).

RESPONSE: This experimental research was performed in 2010. No formal hearing studies were conducted as assessments and inner ear sampling of animals started in 2011. Clinical assessments score of NHPs have been added in the manuscripts supplementary (Table S1) to account for responsiveness/behavior, posture/appearance, neurological function, vocalization and other clinical signs (rash, bleeding, gastrointestinal (GI) symptoms, urine output, edema, respiration, and exudate).  A mean clinical score for all animals have been recorded for each day-post exposure (DPE).   

Fig 2 : Heart rate of the survivor bimodal ?

RESPONSE: The reviewer’s term “bimodal” has been added and the description has been provided for this bimodal heart rate of the survivor as observed in Fig. 1, Fig. S1c, and L525-L533. Wauquier et al. 2020 associates elevated heart rate with cellular signaling events leading to vascular endothelial dysfunction.  This bimodal heart rate and temperature is likely associated with the bimodal cytokines responses (Fig. S9)

L343 : Survivor: remarquable pic alkaline phosphatase and gamma-glutamyl transferase - explain? Fig 5

RESPONSE: Similar to other liver function enzymes (i.e. AST and ALT), alkaline phosphatase (ALKP) and gamma-glutamyl transferase (GGT) are found in many organs throughout the body with the highest concentration found in the liver. L346-L348 indicates levels in the survivor were lower than maximum reference parameters (ALKP<2633 IU, GGT<136 IU/L, Park et al.).  In Figure 5, ALKP and GGT levels of survivor were near base levels during the recovery stage (DPE 30-41).

L414 Fig 8: Remarkable Pathological assessment

RESPONSE: The authors greatly appreciate the kind words about Fig. 8.

L500 "experimental history"

RESPONSE: The term has been added to related discussion section.

L500 also "limited extend" not for human and Cynomolgus model has been studied before...

RESPONSE: Changes have been made for this line, but not in the exact phrase.

L519 and 561 see also " High heart rate at admission as a predictive factor of mortality in hospitalized patients with Lassa fever: An observational cohort study in Sierra Leone. J Infect. 2020 Feb 3. pii: S0163-4453(20)30061"

RESPONSE: This reference is highly valued.  Thank you very much.  As previously discussed, it has been added to describe bimodal heart rate and temperature.

L548 Important discussion "why one survived?" see L 343

RESPONSE: In discussion section, the paragraph related to this line describes why the survivor may have thrived unlike the non-survivors. Survival of the animal is likely linked to the ability for hemoglobin to regulate pH (e.g. bicarbonate and total carbon dioxide). Hemoglobin levels were higher in survivor than non-survivors until DPE 8 during metabolic alkalosis instead of respiratory alkalosis, respectfully.  

Two important point to present/discuss

- At first: The emerging geographic extension of the Lassa fever in West Africa (as well as exported cases) strongly favor of this type of research on LASV

RESPONSE: As referenced in L52-L53, Macher et al. 1990 “Historical Lassa Fever Reports and 30-year Clinical Update” provides a table of import-cases of Lassa fever of people arriving from Western Africa into the U.S., United Kingdom, Germany and several more counties. Air travel is strongly noted for most imported-cases. The population with Lassa fever were mostly medical workers, aid workers, and engineers. Haas et al. 2003 “Imported Lassa Fever in Germany: Surveillance and Management of Contact Persons” describes man returning from Togo who infected, and 232 people were exposed.  No symptomatic secondary infection were observed.  However, a physician tested positive for LASV-IgG when in contact with patient on day 9 of illness.  Ehlkes et al. 2017 “Management of a Lassa fever outbreak, Rhineland-Palatinate, Germany, 2016” describes an aero-evacuated health professional infected an undertaker. Importantly, the rodent known to be the reservoir for LASV (Mastomys natalensis) has been captured in Southern Africa as indicated in Ishii et al. 2011 “Novel Arenavirus, Zambia.”   Added to L53-55; “A primary concern for imported Lassa fever cases are severely infected commercial air travelers and the international transport of infected Mastomys natalensis.”

- L 505/ Neural system (see  Animal Model of Sensorineural Hearing Loss Associated with Lassa Virus Infection. J Virol. 2015 Dec 30;90(6):2920-7. doi: 10.1128/JVI.02948-15.

RESPONSE: The citation is added to L519. This alternative model is a great source to understand LASV induced hearing loss using STAT1 knockout mice.  Also, it is an indirect example of how interferons assist in fighting the virus. However, STAT1 has multiple indirect effects as the transcription factor cooperates and/or antagonizes activities of other transcription factors on hundreds of different gene promoters and related cis-regulatory elements. 

Reviewer 3 Report

            The manuscript by Downs et al. contains a relatively straightforward experiment distinctive for the information it will provide the scientific community in reference to a wide range of measurable parameters in cynomolgus macaques infected with Lassa virus.  The “importance” or “reader interest” in this manuscript is directly proportional to the degree of international interest in mitigating or preventing Lassa fever in humans; thus, as with any hazardous emerging virus (exemplified by SARS-CoV-2), widespread indifference can rapidly change to a search for pathogenesis information of the kind contained here.

            Ordinarily, the lack of paired controls might disqualify this study as too difficult to interpret, but BSL-4 experiments in nonhuman primates (NHP) often rely (for reasons of safety, feasibility, and animal welfare) upon historical controls.  In this respect, the manuscript could achieve more clarity, readability, and credibility in its interpretations if historical findings were discussed in parallel.  Things cited in the abstract (anemia and thrombocytopenia, for example) represent trends that remain within normal bounds, and appear to have little if any pathognomonic or prognostic value; more important, it cannot be clear to a reader whether these result more from procedures, and less from viral disease. If controls cannot be run simultaneously, it would be helpful to reference historical normal controls, as well as similar experiments (using telemetry or not) with viruses the have no vascular or other multi-organ consequences in cynomolgus macaques.

            I do not fault the experiment for being “merely descriptive,” as high-quality descriptions of pathogenetic events can be helpful for later experiments, and should be made accessible. However, the introduction, discussion, and even results tend to lack context, making them useful only to those who already know all the prior Lassa fever literature, as well as knowing where misleading results can be obtained in similar NHP experiments.  In short, I ask the seniormost team members to be more mindful of readers in explaining what is truly new and useful in this manuscript.

            In much of the narrative, the one survivor (of four animals) is offered as a kind of control, but this was an unplanned event that does not appear to hinge on any of the measured parameters, and provides no meaningful information on the pivotal events (days 8-10 post-infection) that could contribute to survival.  There are four experimental animals and no control; narrative needs to explain more fully which results are meaningful, and why.

            Another curious absence in this manuscript is a comparison with pathogenesis observed with aerosol infection (here) and parenteral infections (many historical manuscripts) in cynomolgus macaques. Whether vaccines or therapies have identical or disparate effects in subjects infected by parenteral or aerosol routes, it would be useful here to note any pathogenetic differences observed.  Otherwise, was there any point to using aerosol infection?

            In technical matters, I am especially puzzled by Figure 7:a-c.  How can classical and activated monocytes be 25-45% positive for Lassa antigen on day 0? (certainly not from the administered dose of virus). Is this an experimental artifact of Fc receptors in the assay?  The “-fold increase” is an unhelpful and seemingly misleading way to try to make meaning of these results.  Perhaps I am missing something, and missed its explanation as well.  Or perhaps indecipherable results should be moved to Supplemental if they are to be included at all.

            Minor technical/presentation point. Authors to their credit note the technical challenges and note procedures “On days where sampling via CVC was complicated due to kinking 115 of the line…”.  Can authors assure readers that these unplanned samplings do not explain any of the outlying data points?  Can some indication (asterisk?) be used to indicate samples taken under anesthesia?

            There are some minor errors possibly introduced by a rogue spell-checker (e.g. Line 170 “anti-mouse polymer”).  I think semicolons are often used wrongly, but meaning isn’t significantly affected, so maybe OK with a modern editor.

Author Response

  First and foremost, the authors of this manuscript greatly appreciate the significant contribution made to review this article and provide feedback to make improvements.  Thank you very much.  Below, are responses to comments (REVIEWER 3 Comments in italics).

            Ordinarily, the lack of paired controls might disqualify this study as too difficult to interpret, but BSL-4 experiments in nonhuman primates (NHP) often rely (for reasons of safety, feasibility, and animal welfare) upon historical controls.  In this respect, the manuscript could achieve more clarity, readability, and credibility in its interpretations if historical findings were discussed in parallel.  Things cited in the abstract (anemia and thrombocytopenia, for example) represent trends that remain within normal bounds, and appear to have little if any pathognomonic or prognostic value; more important, it cannot be clear to a reader whether these result more from procedures, and less from viral disease. If controls cannot be run simultaneously, it would be helpful to reference historical normal controls, as well as similar experiments (using telemetry or not) with viruses the have no vascular or other multi-organ consequences in cynomolgus macaques.

RESPONSE (P1. Sentence 1): The authors greatly appreciate comments regarding limitations working under BSL-4 conditions. This BSL-4 research would benefit in the use of a higher number of non-human primates (e.g. controls). Given this is a natural history study, baseline measurements were compared to day-post exposure (DPE) readings. The goal of using baseline measurements was to determine the status of the animal prior to challenge to identify any predispositions. Baseline readings were obtain using the average value of each animal prior to Lassa virus (LASV) challenge, which were marked by horizontal dashed lines in figures. Baseline telemetric readings of each cyanomolgus macaque are uniquely marked by grey lines. Park et al. 2016 “Reference values of clinical pathology parameters in cynomolgus monkeys (Macaca fascicularis) used in preclinical studies” provides reference values that are incorporated in the current manuscript. Baseline values of each animal were within the range of preclinical reference values.

RESPONSE (P1. Sentence 2): Based on current recommended changes, historic findings are discussed in the introduction (L60-L69), results (L312-L323, L336-L337, L344-L350, L368-L369, L377-L383, L386-L387, L410-L411) and discussion sections (L509-L515, L528-L533, L539-L541, L544-L548, L554, L561-L563) of the manuscript.

RESPONSE (P1. Sentence 3): Changes have been made in the result section relative to the previous responses. In accordance to Fortman et al. 2018, temperature and heart rate are dependent on method and circumstances under which they are measured. Telemetric data for fever was based on color-coded temperature readings > 1.5°C over baseline (Fig. S1-S2). Color-coded heart rate, mean arterial blood pressure (MAP), and respiratory rate identified baseline, values significantly higher: >3.0 SD from corresponding baseline, and values significantly lower: < 3.0 SD from corresponding baseline.  These significant changes help determine hypotension, and tachypnea, respectfully. In accordance to Fortman et al. 2018, hypotension was determine using baseline standard deviation and tachpnea was values ≥ 54 respirations per minute. As described in P1. Sentence 1, reference values from Park et al. 2016 help describe other clinical parameters and dashed lines in graphs visually indicate baseline averages before LASV challenge.

  Lymphocytopenia was determine with when values were ≤ 1,870 cell/µl (DPE 5; Fig 6i, 6l). As recommended monocytopenia has been remove from the manuscript. L368-L369 contains: “In accordance with Adams et al. 2014, anemia was identified in all animals on DPE 8 with hemoglobin ≤ 10.4 g/dL and hematocrit percentage ≤ 34.8% (Fig. 6c-6d).” On L376-L379, thrombocytopenia is addressed using Park et al. 2014; “All animals had thrombocytopenia by DPE 8 as the numbers of platelets were less than lower limits (≤ 3.32X105/µl; Fig. 6e).” As previously described, alkalosis is determine with blood pH > 7.5. Hypoproteinemia could be described, in L342-L343, with the significant drop in total protein (Fig. 5a) and albumin (Fig. 5b) relative to reference values in Park et al. 2014.

Response (P1. Sentence 4): Refer to P1. Sentence 1.

            I do not fault the experiment for being “merely descriptive,” as high-quality descriptions of pathogenetic events can be helpful for later experiments, and should be made accessible. However, the introduction, discussion, and even results tend to lack context, making them useful only to those who already know all the prior Lassa fever literature, as well as knowing where misleading results can be obtained in similar NHP experiments.  In short, I ask the seniormost team members to be more mindful of readers in explaining what is truly new and useful in this manuscript.

RESPONSE (P2): All of the reviewers’ feedback provide an outside point of view from experts that increase the value of the manuscript.  This feedback assists with updating the clinical observations seen in Table S1 and Fig. 1 disease progression timeline of the aerosol model.  This timeline and recent changes will help explain the order of events that can be used to strategize future intent of health science researchers. 

            In much of the narrative, the one survivor (of four animals) is offered as a kind of control, but this was an unplanned event that does not appear to hinge on any of the measured parameters, and provides no meaningful information on the pivotal events (days 8-10 post-infection) that could contribute to survival.  There are four experimental animals and no control; narrative needs to explain more fully which results are meaningful, and why.

RESPONSE (P3. Sentence 1): Refer to P1. Sentence 1 relative to baseline controls of four experimental animals and reference values. For determining what contributed to survival, the clinical and laboratory parameters have been described in L564-L570 as well as clinical assessment (Table S1) has been added to the supplementary material. The extensive amount of data provided in Fig. 2-8, Table S1, and Figure S1-S10 are summarized in Fig. 1.  

RESPONSE (P3. Sentence 2): The following comment is added to L53-55: “A primary concern for imported Lassa fever cases are severely infected commercial air travelers and the international transport of infected Mastomys natalensis.”  This natural history study, relative to the FDA definition on L70-L73, has the potential to improve countermeasures serving as an animal model for vaccine development, improve awareness of deliberate aerosol use, and identifies disease progression in Fig. 1.  Using the reviewers’ feedback, this meaningful content has been included in the abstract.

            Another curious absence in this manuscript is a comparison with pathogenesis observed with aerosol infection (here) and parenteral infections (many historical manuscripts) in cynomolgus macaques. Whether vaccines or therapies have identical or disparate effects in subjects infected by parenteral or aerosol routes, it would be useful here to note any pathogenetic differences observed.  Otherwise, was there any point to using aerosol infection?

RESPONSE (P4): The goal was to examine the aerosol route due to its role as a potential biothreat and for developing related medical countermeasures. This study provides first major study describing the aerosol route in greater extent than previous references cited. No profound difference in pathology has been identified between the intramuscular and aerosol exposed route even with the pulmonary lesions, which makes sense for an aerosol challenge.  Most studies do not present this extensive amount of data, so a broad comparison cannot be made to identify major differences in data or even if methods were ever used in LASV studies. To our knowledge, no formal side by side comparison has been performed probably because of the number of variables involved between experiments.  

            In technical matters, I am especially puzzled by Figure 7:a-c.  How can classical and activated monocytes be 25-45% positive for Lassa antigen on day 0? (certainly not from the administered dose of virus). Is this an experimental artifact of Fc receptors in the assay?  The “-fold increase” is an unhelpful and seemingly misleading way to try to make meaning of these results.  Perhaps I am missing something, and missed its explanation as well.  Or perhaps indecipherable results should be moved to Supplemental if they are to be included at all.

RESPONSE (P5. Sentence 1-2): This key point brings to light required changes.  In L408-L409, “Non-specific binding of antibodies were observed as LASV-positive classical and activated monocytes occurred on DPE 0” and further changes are made in L32-L33.

RESPONSE (P5. Sentence 3-5): “Fold is defined by mean of DPE for all animals of interest divided by mean at DPE 0.” is added to L403-404. This approach enabled the identification of change relative to baseline, which is marked by horizontal dashed lines in Fig. S8.

            Minor technical/presentation point. Authors to their credit note the technical challenges and note procedures “On days where sampling via CVC was complicated due to kinking 115 of the line…”.  Can authors assure readers that these unplanned samplings do not explain any of the outlying data points?  Can some indication (asterisk?) be used to indicate samples taken under anesthesia?

RESPONSE (P5): The following statement has been added to L127-L128 “Survivor and D10:Non-survivor had catheter kinking three days prior to challenge; no other kinks are noted” Metabolism of anesthesia and its related effects would pass by the time of challenge three days later.

            There are some minor errors possibly introduced by a rogue spell-checker (e.g. Line 170 “anti-mouse polymer”).  I think semicolons are often used wrongly, but meaning isn’t significantly affected, so maybe OK with a modern editor.

RESPONSE (P6): Thank you for addressing the error. Corrections have been made.